# Unified, Labeled, and Semi-Structured Database of Pre-Processed Mexican Laws

**Bella Martinez-Seis** [1,*], **Obdulia Pichardo-Lagunas** [1,*], **Harlan Koff** [2,3,4], **Miguel Equihua** [5], **Octavio Perez-Maqueo** [5] and **Arturo Hernández-Huerta** [5]

[1] Engineering Department, UPIITA-IPN, Instituto Politécnico Nacional, Mexico City 07360, Mexico
[2] Department of Geography and Spatial Planning, University of Luxembourg, Maison des Sciences Humaines, 11, Porte des Sciences, L-4366 Luxembourg, Luxembourg; harlan.koff@uni.lu
[3] GAMMA-UL Chair for Regional Integration and Sustainability, Instituto de Ecología, A.C. (INECOL), El Haya Xalapa 91070, Mexico
[4] Department of Politics and International Relations, University of Johannesburg, Auckland Park 2006, South Africa
[5] Red de Ambiente y Sustentabilidad, Instituto de Ecología, A.C. (INECOL), Xalapa 91073, Mexico; equihuam@gmail.com (M.E.); octavio.maqueo@inecol.mx (O.P.-M.); arturo.hernandez@inecol.mx (A.H.-H.)
[*] Correspondence: bcmartinez@ipn.mx (B.M.-S.); opichardola@ipn.mx (O.P.-L.)

**Abstract:** This paper presents a corpus of pre-processed Mexican laws for computational tasks. The main contributions are the proposed JSON structure and the methodology used to achieve the semi-structured corpus with the selected algorithms. Law PDF documents were transformed into plain text, unified by a deconstruction of law–document structure, and labeled with natural language processing techniques considering part of speech (PoS); a process of entity extraction was also performed. The corpus includes the Mexican constitution and the Mexican laws that were collected from the official site in PDF format repealed before 14 October 2021. The collection has 305 documents, including: the Mexican constitution, 289 laws, 8 federal codes, 3 regulations, 2 statutes, 1 decree, and 1 ordinance. The semi-structured database includes the transformation of the set of laws from PDF format to a digital representation in order to facilitate its computational analysis. The documents were migrated to JSON type files to represent internal hierarchical relations. In addition, basic natural language processing techniques were implemented on laws for the identification of part of speech and named entities. The presented data set is mainly useful for text analysis and data science. It could be used for various legislative analysis tasks including: comprehension, interpretation, translation, classification, accessibility, coherence, and searches. Finally, we present some statistic of the identified entities and an example of the usefulness of the corpus for environmental laws.

**Keywords:** Mexican legislation; laws; natural language processing; legislative documents

## 1. Introduction

The legislation of a country is the set of laws through which the coexistence of a society is regulated. The study of the different legislative documents can provide information on their evolution and impact; however, extracting the information and analyzing the large number of documents by human means can be complicated and requires time and resources. The automatic analysis of text could help on this task, but in order to fulfill it they have to be machine-readable and formally represented [1].

In Mexico, several computation studies have been conducted on law documents, mainly focused on a particular topic such as: abortion [2,3], violence [4–6], or even computation [7]; however, none of them have been supported by computational algorithms that would facilitate the exploration, search, compilation, analysis, and interpretation of laws. There is a need to create a repository of normative documents into a format that allows computational processing on them and with it the extraction and analysis of relevant data.

The Mexican constitution is the root document for Mexican legislation. It is followed by laws that develop the mandates stated by the constitution and by norms that regulate the implementation of the laws. Even though Mexican law documents tend to have a fairly common structure, it is not always the same due to different cultural and governmental contexts and the publication time of laws. For example, the current Mexican constitution dates back to 1917. To enable computational processing of law documents, it is necessary to develop a strategy to homologate the occurring variations in structure and style. This was a key task while performing the creation of the present corpus. We manually analyze several law documents and deduce a generalized structure of the laws. As a result, we have a hierarchical structure given by levels of the elements of the laws such as chapters and articles—this process is what we call the deconstruction of laws because of the understanding between text and the hierarchical structure inside each law.

The laws were first transformed into plain text. Then, a script with designed rules was applied in order to clean the plain text from noisy elements such as footers and headers. The plain text is transformed into the semi-structured form of the law documents given by the designed hierarchical structure. We pre-process each element of the structure using algorithms in order to obtain the part of speech (PoS), lemmatization, tagging, and named entity recognition (NER).

We present a corpus of pre-processed Mexican laws that could be very useful for text analysis and data science. A lot of studies on the peculiar use of natural language within the legal domain focus on comprehension, interpretation, translation, classification, coherence, searches, and so on. Further, applications for final users could be built in order to access those documents or for legal counsel. The corpus generated in this project seeks, in the first instance, to evaluate the impact of Mexican legislation on the implementation of environmental and sustainability public policies in the country. A preliminary exercise was conducted to produce a statistical analysis based on the generated corpus.

## 2. Methods

To deal with variations in structure among Mexican laws, we analyze the different structures of law documents and we deduce a general structure that comprised the following semantic elements: titles and transient, chapters, sections, and articles. This structure was used to define a JSON format by the deconstruction of the structure of law documents.

We developed a unified semi-structured database in JSON format of Mexican laws. While processing all documents, we conducted, for each element of the database, part of speech (PoS) tagging [8]. This tagging also included entity identification using several natural language processing metrics. Figure 1 shows the methodology used for the creation of the proposed unified semi-structured database of the text content of Mexican laws. The building process started by collecting law documents in PDF format from the official site of the Mexican legislative power. Then, we transformed those documents into plain text using an API and proper rules, which was one of the main challenges. Given the critical importance of that process, we created an algorithm for the cleansing process to systematically apply the proper order of text in tables and to remove headers and page footers.

Data transformation converts all the laws into a unified semi-structured format represented by a JSON structure designed by the team. It contains tags to separate semantic elements and enforce hierarchies within the data. Finally, two natural language processing tasks were applied. One is the entity detection task that extracts entities such as concepts, complete names of persons, places, or laws. The other is the tagging process, which detects

part of speech. Finally, we obtain the term frequency (TF) before and after the entity detection. For each word or entity, we obtain its lemma and its grammatical function as a label of EAGLES standard [9]. The following subsections describe in more detail the methodology used for corpus generation.

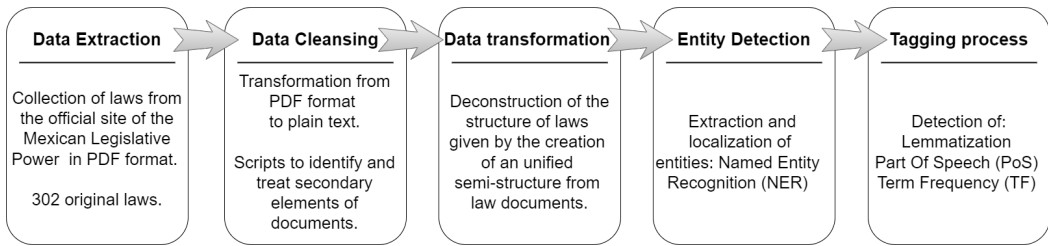

**Figure 1.** Stages for construction of the pre-processed law corpus.

### 2.1. Data Extraction

The root of the legal tree is the constitution, which dates back to 1917 but has been widely amended since. Currently, it has 9 titles, 136 articles, and 19 transitional provisions. On the other hand, there are 294 federal laws in force, all of them develop the general mandates stated in the constitution. There are different types of laws: organic laws (regulating a governmental body), federal laws (regulating the three governmental levels), state/local laws (of each federal entity and according to the particular state constitution), general laws (dealing with overarching/recurring matters), and regulatory laws (following a specific constitutional mandate). A code is a systematic and homogeneous collection of laws regulating a branch of positive law. The corpus includes: the Mexican constitution, 289 laws, 8 federal codes, 3 regulations, 2 statutes, 1 decree, and 1 ordinance, of which, 14 are repealed and the rest are in force. Documents were collected from the official site [10] of the Mexican legislative power in PDF format.

### 2.2. Data Cleansing

Once the set of laws included in the corpus were identified, the components that constituted the content of the document were determined. In order to transform from PDF format to plain text, we evaluated PDFMiner [11], PyPDF2 [12], tabula-py [13], PyDF2JSON [14], and Tika [15]. For the manual evaluation, we consider: special characters, size of the document, pages style, text in images, text in tables, and execution time.

We, finally, used Apache Tika [15] to transform documents into plain text—it is a content analysis and detection framework written in Java that contains a toolkit that detects and extracts metadata and text content in formats such as: Word, Excel, PDF, JPG, or mp4. With Tika we obtained all the text of the document, including text in tables and images; the headers and footers repeated as many times as there are pages. Python scripts were implemented to identify and treat: headers and footers, tables, images, page numbers, and modification notes, keeping only the texts. It was the most challenging step because of the diversity of documents and the special characters in Spanish. In order to solve it, special scripts were designed to:

- Identify the headers and footers of each document.
- Generate a list of each document with its respective header.
- Regular expressions were used for footers which include repetitive text and page number.

    Further, manual validation of the text was performed by a specialist in NLP and lawyers.

### 2.3. Deconstruction of the Structure of Laws

We carried out a study of the internal hierarchical structure of the laws in order to obtain a representation of the document that would reduce the loss of information as much as possible and, on the contrary, enhance the data that could be obtained from the documents. The analysis showed that laws do not maintain a uniform structure. We manually analyze several law documents and deduce a generalized structure of the laws. By

the understanding between text and the organization inside each document, the hierarchical structure given in JSON format was generated as a deconstruction of the document.

In this structure, all the possible fields that the document could contain, or not, were integrated and unified in JSON type files. The above facilitates the representation of data by levels, in such a way that it can be interpreted that titles contain chapters, which in turn contain articles. Figure 2 shows the general structure of that hierarchy.

**Figure 2.** Proposed deconstruction of the internal hierarchy of laws divisions (titles, chapters, articles, etc.) for document representation.

Some challenges for the automatic transformation from plain text to the generalized structure in JSON were:

- Lack of homogenization of document structure not only by time but also by the same government. Some laws lacks of sections or include them in a different hierarchical order.
- Spelling mistakes such as orthographic accents and abbreviations.
- The numbering of the same element can be cardinal or ordinal numbers.
- Special articles as repealed articles and transitory articles, which are articles used when new regulations emerge or change; therefore, they are effective during the transition.

The generated structure was evaluated, analyzed, and manual validated by specialist in NLP, Mexican lawyers, and experts in public policies in Mexico and other countries (South Africa, Finland, and Luxembourg).

Regarding the structure: each law has a name and a general description. The laws have divisions where title and transient articles were considered as the second level. Each title has a number and name. Chapters are usually inside the title, but sometimes would not be. Inside the chapter, we can find articles that are the fundamental division of laws. Each of the elements of this hierarchical deconstruction of laws is described in greater detail below.

- **Current Text** *(Texto Vigente)*. General description included in the original law such as: date of publication of last reform, name of the president, side item such as headers, and description in text of seals on covers or margin.
- **Law name** *(Nombre)* Each law has a name. In very few cases, the number and name of the book will be also found in this section; since only very extensive laws are divided into books that try to collect a single branch of law, in this case there will be a document for each book of the law.
- **Titles** *(Títulos)*. Some laws are divided into titles with clearly differentiated parts. This division is presented in extensive and general laws.
    - **Title** *i (Título i)*. This section has the word *Title* and its ordinal number.
    - **Title Name** *(Nombre del título)*. This tag has the name assigned to the title.
        * **Chapter** *i (Capítulo)*. It is the most common division on laws. In general, chapters could or could not be inside titles. They are numbered with Roman numerals.
        * **Chapter Name** *(Nombre del capítulo)*. Each chapter has a short name.
            · **Article** *k (**Artículo** k)*. The article is the elementary and fundamental division of the laws; it includes a legal provision condensed in a single

or several sentences, sometimes divided into several paragraphs. Each article must regulate a single topic or precept. In this deconstruction, the article tag includes all the paragraphs *(párrafos)*, sections *(apartados)*, fractions *(fracciones)*, and subsections *(incisos)* of that article.

- **Transient *(Transitorios)*.** Provision intended to govern temporary situations that exist prior to the effective date of a law.

### 2.4. Entity Detection

We include an entity identification stage in the pre-processing of these documents. Named entity recognition (NER), also known as entity extraction, consists of locating and classifying parts of the studied text in pre-established categories such as places, people, organizations, expressions of time, and quantities.

To perform the entity recognition process, we use spaCy [16]. This is a free, open-source library for advanced natural language processing (NLP) in Python. spaCy provides a variety of linguistic annotations about a text's grammatical structure—this includes the word types, the parts of speech, morphological analysis, lemmatization, named entity recognition, and dependency parsing.

spaCy works with convolutional neural networks and provides pre-trained models of different languages; in addition, it allows you to create new models or retrain the models you provide with your own data to create models in specific fields. spaCy can recognize various types of named entities in a document, by asking the model for a prediction. The models are statistical and strongly depend on the examples they were trained on. spaCy has a base of named entities in different languages, including Spanish. The NER module for Spanish is trained with the AnCora [17] and WikiNER [18] corpora. Further, spaCy NER model uses capitalization as one of the keys to identify named entities so it is important to use NER before the usual normalization or derivation preprocessing steps.

We look for entities in the text and replace them as one term. For example, Federal Judiciary (*Poder judicial de la Federación*) is an entity because it represents an instance—the entity has five words in Spanish, and its entity might be *poder_judicial_de_la_federación*.

The official name of the country of Mexico is Estados Unidos Mexicanos, such that estados_unidos_mexicanos was the entity that appears in all the documents analyzed. The Official Gazette of the Federation (*Diario Oficial de la Federación*) is the government organ in charge of publishing laws, decrees, regulations, agreements, circulars, orders, and other acts issued by the Powers of the Federation; the entity diario_oficial_de_la_federación appears in all documents except the constitution, the Law of Internal Security (*Ley de Seguridad Interior*), and the law that creates the Autonomous University of Chapingo (*Ley que crea la Universidad Autónoma de Chapingo*). The previous identification of entities allowed the tagging process to be more efficient, since it avoided the separation into words of entities with compound names.

A subset of documents focus on the environment were selected (the selected documents are in Section 4.3). Among those documents, there are 5718 different tokens, of which, 1464 are entities. In total, 43.18% are persons, 36% are organizations, and the remainder is divided between time expressions, places, locations, and quantities. By a manual validation, we detect a few wrong entities—the accuracy is 98.04%, which is similar to what the spaCy algorithm reports. The errors occur in two main cases: The first one is that Roman numerals are commonly considered as the first word of an entity, for example, it detects "iii_the" *(iii_El)* as an entity. The second one is associated with the third person in plural in the present and present subjunctive, as sometimes spaCy includes the verb in this tense into the entity; for example, it detects "contravene_international_law" *(contravengan_el_derecho_internacional)* as an entity.

### 2.5. Lemmatization and Tagging Process

The lemmatization process allows obtaining the canonical form of a set of morphological variants. Our project used FreeLing for the lemmatization process—this tool includes

linguistic dictionaries for Spanish and other languages. These dictionaries are obtained from different open-source external projects. The Spanish dictionary contains over 555,000 forms corresponding to more than 76.000 lemma–PoS combinations. The library includes the affix analysis module, which is able to detect enclitic pronoun verbal forms and diminutive/augmentative suffixed forms. Further, a probabilistic suffix-based guesser of word categories is used for words not found in the dictionary [19].

The tagging process includes the pre-processing of each element of the structure using algorithms in order to obtain lemmatization, part-of-speech (POS) tagging, and term frequency (TF).

Part-of-speech (POS) tagging is a natural language processing process, which refers to categorizing words in a text (corpus) in correspondence with a particular part of speech, depending on the definition of the word and its context. The tagging process could be carried out using spaCy library or Freeling. Both libraries reach an accuracy of 97% [20,21]; it was considered that FreeLing is a tool originally developed for Spanish and that, according to Orellana et al. [22], although the difference is minimal, FreeLing has better execution times. Considering that this task was performed with a large number of documents, minimizing the execution time is a task that must be considered. The tagging process in this project was implemented using FreeLing [19]. FreeLing is a C++ library providing language analysis functionalities, such as morphological analysis, named entity detection, PoS-tagging, parsing, word sense disambiguation, semantic role labeling, etc., for different languages (English, Spanish, Portuguese, Italian, French, German, Russian, Catalan, Galician, Croatian, Slovene, etc.) released under the Affero GNU General Public License of the Free Software Foundation. We used FreeLing for lemmatization and part-of-speech (PoS).

The tags given by FreeLing are the EAGLES standard (Expert Advisory Group on Language Engineering Standards) [23] and includes several codes for adjectives (**A**), adverbs (**R**), articles (**T**), determiners (**D**), nouns (**N**), verbs (**V**), pronouns (**P**), conjunctions (**C**), numerals (**M**), interjections (**I**), abbreviations (**Y**), prepositions (**S**), and punctuation marks (**F**). Table 1 shows an example for nouns, which includes type, gender, number, and others.

**Table 1.** Label description of the EAGLES standard.

| Pos | Attribute | Value | Code |
|---|---|---|---|
| 1 | Category | Noun | N |
| 2 | Type | Common<br>Proper | C<br>P |
| 3 | Gender | Male<br>Female<br>Common | M<br>F<br>C |
| 4 | Number | Singular<br>Plural<br>Invariable | S<br>P<br>N |
| 5 | Case | - | 0 |
| 6 | Semantic Gender | - | 0 |
| 7 | Degree | Appreciative | A |

A manual validation of the tagging process given by FreeLing was performed in the General Law on Ecological Balance and Environmental Protection (*Ley General del Equilibrio Ecológico y la Protección al Ambiente* LGEEPA) by a specialist in library science and expert in document classification and research. There are 2342 tokens, 108 were misclassified, giving an error rate of 4.61%. The minor misclassifications in the tagging process are related to: dates written as numbers was classified as numbers instead of as dates, especially when they are written with diagonals "/", and Roman numerals were detected as substantives

rather than as numbers—because the NER algorithm joins the Roman numeral with the following noun.

The term frequency is the measurement of how frequently a word or entity occurs in a document. We calculate the term frequency (TF) for each document before and after the entity detection. Then, for each element of the structure (title, chapter, or article) a list of words/entities is given with its respective TF. There is a TF for the original content and a second TF after entity identification.

## 3. Data Description

Original laws were collected in PDF format from the official site of the Mexican legislative power. We transformed them into text using Python scripts and then we cleaned, analyzed, and tagged part of speech with implemented natural language algorithms.

After the entity detection and tagging process, each current text, law name, title name, chapter name, transient content, and article content has:

- Original text or content.
- Term frequency (TF) of the words in the original text.
- PoS tagging of the content.
- Original text with entity replacement.
- Term Frequency of text with entity replacement.

Those elements were place in semi-structured data bases in JSON format using the hierarchy of Figure 3.

```
Δ Content (Contenido)
Δ TF of content (Frecuencia de palabras)
  ▲ Word : tf (Palabra: tf)
Δ PoS tagging of content (Etiquetado)
Δ Entities (Entidades)
Δ TF of entities (TF de entidades)
  ▲ Word : tf (Palabra: tf)
```

**Figure 3.** Hierarchy used for the JSON file of the NLP elements.

In order to process all the laws, a parallel execution was made in a supercomputer because of the execution time. We used the Cuetlaxcoapan supercomputer of the Benemérita Universidad Autónoma de Puebla (BUAP), it is made up of 136 standard computing nodes, 6 special nodes, and a 520 TB storage system, interconnected in an Infiniband network with a star topology. The basic execution unit is a processing core. Each standard compute node has 24 cores (2 Intel Xeon E5-2680 v3 sockets at 2.5 Ghz with 12 cores per socket) and 128 GB of shared RAM; therefore, computation time is measured in core hours. The library "multiprocessing" was implemented, which is a package that offers both local and remote concurrency, effectively side-stepping the global interpreter lock by using subprocesses instead of threads. Due to this, the multiprocessing module allows us to fully leverage multiple processors on a given machine [24]. Documentation processing in this system used a total of 480 core hours.

It allows us to extract the identified features. For each law, two JSON documents are constructed for the semi-structured data base.

Using the deconstruction of the structure of laws (Figure 2) and the elements of the tagging process with NLP (Figure 3), the two final JSON files for each law have the structure presented in Figures 4 and 5, which correspond to the three JSON files for each document in the semi-structured database.

Figure 4 shows the hierarchy of the text database presented in JSON format. It has the deconstruction of the structure of laws (Section 2.3) and the grammar tagging of the content (Section 2.5). It includes the original term found in the text, its lemma, its identification label according to the EAGLES standard, and the probability of success regarding the label.

Figure 5 shows the JSON structure with term frequency for each document. Figure 6 shows the JSON structure that mixes the deconstruction of the structure of laws with the entity detection process and the calculation of the term frequency for each document.

The information (tag) provided for each word or entity is the following: original word found in the text, lemma, identification label according to the EAGLES standard and finally the probability of success regarding the label.

```
– Current text (Texto vigente)
    Δ Content (Contenido)
    Δ PoS tagging of content (Etiquetado)
– Law Name (Nombre)
    Δ PoS tagging of law name (Etiquetado)
– Titles
    • Title ι (Título con número ordinal ι)
        ○ Title Name (Nombre del título)
        Δ PoS tagging of content(Etiquetado)
            ▪ Chapter j (Capítulo con número romano j)
                ▫ Chapter name (Nombre del capítulo)
                Δ PoS tagging of content (Etiquetado)
                    ◆ Article k (Artículo número k)
                        ◇ Article content (Contenido del artículo)
                        Δ PoS tagging of content(Etiquetado)
    • Transient
        ○ Content (Contenido)
        Δ PoS tagging of content (Etiquetado)
```

**Figure 4.** JSON structure for each law with grammar tagging.

```
– Current text (Texto vigente)
    Δ Content (Contenido)
    Δ Entities (Entidades)
    Δ TF of entities (TF de entidades)
        ▲ Word : tf (Palabra: tf)
– Law Name (Nombre)
    Δ Entities (Entidades)
    Δ TF of entities (TF de entidades)
        ▲ Word : tf (Palabra: tf)
– Titles
    • Title ι (Título con número ordinal ι)
        ○ Title Name (Nombre del título)
        Δ Entities (Entidades)
        Δ TF of entities (TF de entidades)
            ▲ Word : tf (Palabra: tf)
            ▪ Chapter j (Capítulo con número romano j)
                ▫ Chapter name (Nombre del capítulo)
                Δ Entities (Entidades)
                Δ TF of entities (TF de entidades)
                    ▲ Word : tf (Palabra: tf)
                    ◆ Article k (Artículo número k)
                        ◇ Article content (Contenido del artículo)
                        Δ Entities (Entidades)
                        Δ TF of entities (TF de entidades)
                            ▲ Word : tf (Palabra: tf)
    • Transient
        ○ Content (Contenido)
        Δ Entities (Entidades)
        Δ TF of entities (TF de entidades)
            ▲ Word : tf (Palabra: tf)
```

**Figure 5.** JSON structure for each law with term frequencies from the original text.

```
– Current text (Texto vigente)
  Δ Content (Contenido)
  Δ TF of content (Frecuencia de palabras)
    ▲ Word : tf (Palabra: tf)
– Law Name (Nombre)
  Δ TF of content (Frecuencia de palabras)
    ▲ Word : tf (Palabra: tf)
– Titles
  • Title i (Título con número ordinal i)
    ○ Title Name (Nombre del título)
    Δ TF of content (Frecuencia de palabras)
      ▲ Word : tf (Palabra: tf)
    ▪ Chapter j (Capítulo con número romano j)
      ▫ Chapter name (Nombre del capítulo)
      Δ TF of content (Frecuencia de palabras)
        ▲ Word : tf (Palabra: tf)
        ◆ Article k (Artículo número k)
          ◇ Article content (Contenido del artículo)
          Δ TF of content (Frecuencia de palabras)
            ▲ Word : tf (Palabra: tf)
  • Transient
    ○ Content (Contenido)
    Δ TF of content (Frecuencia de palabras)
      ▲ Word : tf (Palabra: tf)
```

**Figure 6.** JSON structure for each law with term frequencies over the text with entities substitution.

Table 2 shows some examples of entities and words extracted from the constitution. It should be noted that the labels were assigned to the terms in the Spanish language. E1 (Example 1) has the word resolutions whose motto is its singular resolution; its label classifies it in the category of noun (**N**), of the type common (**C**), with feminine (**F**) gender, the attribute number expresses that it is a plural (**P**), with the attributes case; semantic gender and degree not specified so the value is 0. E2 has an entity detected in the previous stage instead of a word; its label classifies it as a proper noun (**NP**). E3 is an adjective (**A**) and the with a probability of success regarding the label is 0.99295. E4 and E5 are verbs (**V**).

**Table 2.** Example of tags on words or entities identified in the constitution. Each tag has: lemma, label, and probability of success regarding the label.

|  | **Word/Entity** | **Lemma** | **Label** | **Probability** |
|---|---|---|---|---|
| E1 | resolutions (*resoluciones*) | resolution (*resolución*) | NCFP000 | 1 |
| E2 | District_Judges (*Jueces_de_Distrito*) | district_judges (*jueces_de_distrito*) | NP00SP0 | 1 |
| E3 | fundamental (*fundamentales*) | fundamental (*fundamental*) | AQ0CP00 | 0.99295 |
| E4 | issued (*expedidas*) | issue (*expedir*) | VMP00PF | 1 |
| E5 | start (*inicie*) | start(*iniciar*) | VMSP3S0 | 0.424658 |

## 4. Introductory Statistical Analysis of the Corpus for Future Directions

The presented corpus of pre-processed Mexican laws is mainly useful for computational process focus on text analysis and data science. It could be used for several legislative analysis tasks including: comprehension, interpretation, translation, classification, accessibility, coherence, and searches.

A total of 301 documents were transformed into the proposed structures. There were 27,404 articles plus the transient articles, an average of 91.95 articles per document and standard deviation of 105.37 because there are laws with two articles and others with 1010 articles plus their transient articles.

### 4.1. Corpus Statistics

Among the law documents, 53,274 different tokens (entities or words) were identified. The Federal Labor Law presented 11,977 different entities or words discarding the number of repeated appearances, and the General Health Law with 10096 entities or words. The Mexican constitution has 4499 different tokens, remaining among the first 20 laws with more entities or words.

There are 28,158 tokens in just one document, by discarding those, Figure 7 shows the ratio between the number of tokens in each law and the number of laws with a token. We can see that there are a lot of entities or words concentrated in a few documents and just a few entities or words that are in most of the documents. An entity or word appears on average in 6.68 documents with a standard deviation of 20.34, which translates into a higher standard error of the mean. This is because there are few entities or words that are used in many documents, while there are many entities or words that are used in a single document.

Considering only the entities or words identified from the total number of documents, a behavior similar to Zipf's Law is observed, even when not all the vocabulary is considered.

### 4.2. Term Frequency Statistical Analysis

From the term frequency (TF) given to entities and words, a deep analysis of the presence or absence of key terms could be performed. In this sense, we searched for the occurrence of terms in all the corpus documents, considering that if a term is used several times in a document, it is only counted as one occurrence. For example, the term "fight corruption" *(combate a la corrupción)* is in one document but the term "anti-corruption" *(anticorrupción)* detected in the substitution of synonyms phase is in two more documents; its antonym, the term "corruption" *(corrupción)* is used in 14 different documents. Another example is the word "inclusion" *(inclusion)*, which appears in 12 documents, while gender equity *(equidad de género)* is only mentioned in two; this may be due to the fact that gender equity is a recent term and that inclusion is not only focused in terms of gender. In the previous particular examples, we found some disconcerting results to us. Perhaps they might be perfectly understandable for experts in legislation, politics, economy, and environment, but then again they probably also demonstrate a perspective that is hard to appreciate.

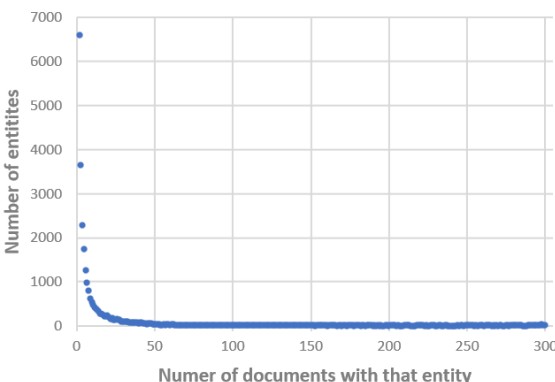

**Figure 7.** Relation of the number of laws in which a certain number of entities or words appear.

Traditionally, specialists in public policies and laws compare laws, regulations, programs, and plans with national and international goals. They have to read a lot and look for keywords in order to evaluate and apply the normative; the interpretation of the law is a key element to establish the meaning and scope of legal norms. The presence or absence of terms/entities should be studied in greater depth in several laws. Irrespective of the interpretation, with the computational corpus ready, those specialists can focus on their analytic insights, instead of expending a very large amount of time and effort to the huge task of cross-reading all the laws.

### 4.3. Inverse Document Frequency (IDF) Analysis for Environmental Laws

There are many kinds of algorithms that can be used to obtain the main ideas or keywords in a document. One of them is TF-IDF (term frequency-inverse document frequency), which is one of the most recognized word weighting algorithms. We concentrated for this case study on the eight main environmental laws:

1. **LGEEPA:** General Law on Ecological Balance and Environmental Protection *(Ley General del Equilibrio Ecológico y la Protección al Ambiente).*
2. **LAN:** National Water Law *(Ley de Aguas Nacionales).*
3. **LDRS:** Sustainable Rural Development Law *(Ley de Desarrollo Rural Sustentable).*
4. **LGPAS:** General Law for Sustainable Fisheries and Aquaculture *(Ley General de Pesca y Acuacultura Sustentables).*
5. **LGVS:** General Wildlife Law *(Ley General de Vida Silvestre).*
6. **LGDFS:** General Law of Sustainable Forestry Development *(Ley General de Desarrollo Forestal Sustentable).*
7. **LGPGIR:** General Law for the Prevention and Integral Management of Waste *(Ley General Para la Prevención y Gestión Integral de Residuos).*
8. **LFBOGB:** Federal Law on Biosafety of Genetically Modified Organisms *(Ley Federal de Bioseguridad de Organismos Genéticamente Modificados).*

Studying the entities within these laws, we obtain the TF of each document and compare it to the TF-IDF of the eight documents; as a result, we can identify the main representative words of each document, even if we evaluate it against a document in the same environmental field. Figure 8 shows an example for the General Law on Sustainable Fisheries and Aquaculture *(Ley General de Pesca y Acuacultura Sustentable)*—the darker blue dots are terms that distinguish the document from the other laws even though they are all of an environmental nature.

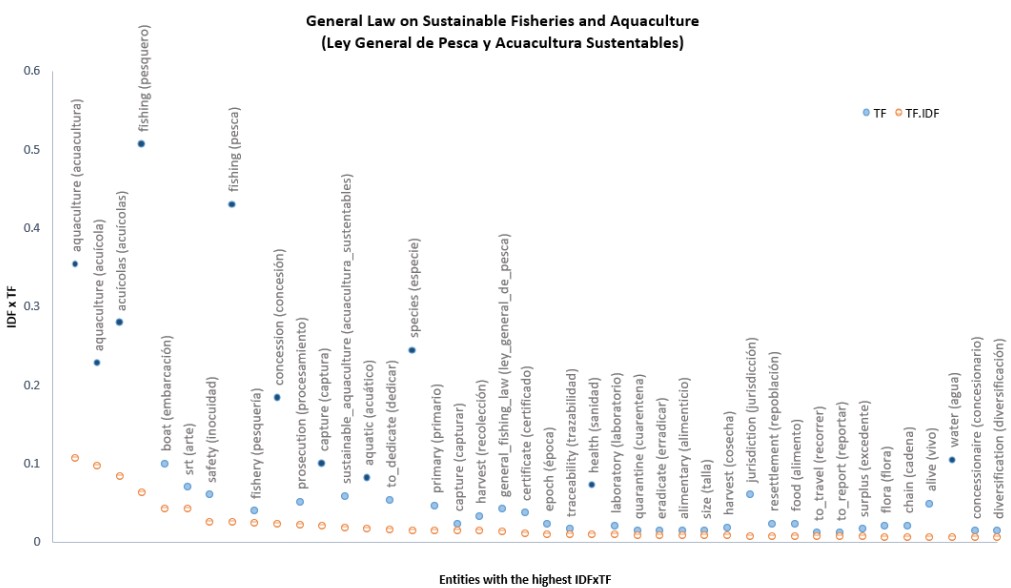

**Figure 8.** Entities with the highest IDF × TF in the General Law on Sustainable Fisheries and Aquaculture *(Ley General de Pesca y Acuacultura Sustentables)* given among eight environmental laws.

Deeper statistical analysis could be performed based on the data set with the computational prepossessing presented in this paper. Further, an artificial intelligence algorithm could be used mainly for natural language processing.

## 5. Conclusions

The corpus described in this document allowed text mining analysis to be carried out on several issues; however, the corpus was not specialized on this subject matter. The corpus is suitable for general use and thus, evaluations can be carried out for any area of interest looking for information from the normative documents in Mexico; therefore, we assert the relevance of producing a data corpus, processed and stored using some kind of computational representation that facilitates the extraction of information from legislative documents. As it can be seen, there is a very wide range of perspectives that can be used to query the corpus.

**Author Contributions:** Conceptualization, M.E. and O.P.-L.; methodology, B.M.-S. and O.P.-L.; software, B.M.-S. and O.P.-L.; validation, M.E., H.K. and O.P.-M.; formal analysis, O.P.-L., B.M.-S., H.K., M.E. and O.P.-M.; investigation, H.K.; resources, O.P.-M. and A.H.-H.; data curation, O.P.-L. and B.M.-S.; writing B.M.-S. and O.P.-L.; review and editing, M.E., H.K. and O.P.-M.; visualization, B.M.-S.; project administration, M.E. and O.P.-L. All authors have read and agreed to the published version of the manuscript.

**Funding:** This research received no external funding.

**Institutional Review Board Statement:** Not applicable.

**Informed Consent Statement:** Not applicable.

**Data Availability Statement:** Site: https://cloud.upiita.ipn.mx/docentes/s/EgY4sQNmZNQ9iNx (accessed on 15 May 2022). Password: IPN_MxLaws_BM_OP.

**Acknowledgments:** This work was part of the general initiative i-Gamma, developed to foster the use of environmental Big data and machine learning to support sustainable development. This collaboration is entitled in the project "*Uso del big data para la gestión ambiental del desarrollo sostenible (Integralidad Gamma)*", FORDECYT-296842. The specialized computational infrastructure was provided by the "*Laboratorio Nacional de Supercómputo del Sureste de México*" laboratory, thanks to the approved project 201903092N. Special thanks to the lawyers and specialists in blibliteconomics for the validations, under the direction of Daniel Francisco Rivera and Zamná Pichardo. We would also like to thank Miguel Hidalgo and Victor Carrera as experts in Data Minig and Natural Language Processing. And our recognition for the public policy team: Julia Ros Cuellar, Jorge-Israel Portillo, Antony Challenger, and Maria-del-Socorro Lara. This article has been supported by the Instituto Politécnico Nacional through the project "20221158-Integración, visualización y proyección para seguimiento del cáncer en México implementando técnicas de Ciencia de datos e Inteligencia artificial." and the research project "20221676-Modelado y generación de enunciados en Lengua de Señas Mexicana usando Redes Neuronales Artificiales".

**Conflicts of Interest:** The authors declare no conflict of interest.

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
