# Peer review of "Unified, Labeled, and Semi-Structured Database of Pre-Processed Mexican Laws"

_data, 2022_

Round 1

Reviewer 1 Report

This paper presented a corpus that allowed text mining analysis to be carried out on several issues. The authors also claim that the corpus is suitable for general use and evaluations can be carried out for any area of interest looking for information from the normative documents in Mexico. Overall, the idea and tool presented in this paper is interesting and the authors give a decent description to the methods applied. Then, the authors presented a case study of ecosystem integrity to demonstrate the usefulness of the corpus. The results discussion is relatively clear.

The main problem with the current paper is that much information is missing, details should be given in many places. More cases will need to be studied in order to demonstrate the usefulness and support the authors’ conclusions. The following are the suggestions from the reviewer.

In the Introduction section, the author described the background and motivation. However, the methods applied in this work should also be mentioned and discussed, previous related literatures should be studied. Moreover, the authors should stress the significance of this work in this section as well.

In the Methods section, line 58 and line 91, the authors specifically mentioned that transformed documents into plain text using an API and proper rules is one of the main challenges. However, little discussion about this challenge is given, the author should discuss the reasons and solutions.

Line 98, “a model was generated in order to deconstruct the document”, can you give more details about this ‘model’? it’s critical to understand how to deconstruct the document.

Line 137, “We used the Python Spacy library, which has a base of named entities in different languages, including Spanish.” This Python Spacy library should be referenced and discussed. Also, many libraries and models applied in this paper should be referenced to help understand the work.

In Section 2.5, a few references are missing.

Line 180, about the parallel execution conducted on a supercomputer, more details should be given. Includes: parallel library, HPC spec, number of CPU/GPUs, time-to-solution, etc.

In Section 4, the authors did a deep analysis of the presence or absence of key terms. However, the reviewer feel not clear how the authors reached to the conclusion that “Irrespective of the interpretation, with the computational corpus ready, those specialists can focus on their analytic insights, instead of expending a very large amount of time and effort to the huge task of cross-reading all the laws.” Perhaps, more thorough discussion should be given. Also, the reviewer suggest adopt a second case to support the claim and show the function.

Author Response

In the Introduction section, the author described the background and motivation. However, the methods applied in this work should also be mentioned and discussed, previous related literatures should be studied. Moreover, the authors should stress the significance of this work in this section as well. 

 A: A brief description of the methodology and methods used was added in the introduction (lines 44 to 49). Previous works are mentioned in line 25. 

In the Methods section, line 58 and line 91, the authors specifically mentioned that transformed documents into plain text using an API and proper rules is one of the main challenges. However, little discussion about this challenge is given, the author should discuss the reasons and solutions. 

A: In the Methods section, the different types of documents that were processed and how the individual characteristics had to be integrated to generate the JSON representation model are mentioned (Line 88). The subsection "Data Cleansing" lists the elements that generated some kind of exception or caused noise when exporting to plain text (Line 107, previously line 91). 

Line 98, “a model was generated in order to deconstruct the document”, can you give more details about this ‘model’? it’s critical to understand how to deconstruct the document. 

A: In the section "Deconstruction of the structure of laws" you can see the elements that compose the JSON representation model as well as its general description. We add more details about the proposed deconstruction (Lines: 40-43, 118-124, 129-139) 

Line 137, “We used the Python Spacy library, which has a base of named entities in different languages, including Spanish.” This Python Spacy library should be referenced and discussed. Also, many libraries and models applied in this paper should be referenced to help understand the work. 

A: The "Entity Detection" section describes the tool used for named entity detection (NER). The reference corresponding to spaCy and Freeling, which were the libraries used for the morphological analysis, was included. (Lines 174, 183-187) 

In Section 2.5, a few references are missing. 

A: References were included. 

Line 180, about the parallel execution conducted on a supercomputer, more details should be given. Includes: parallel library, HPC spec, number of CPU/GPUs, time-to-solution, etc. 

A: The "Data description" section describes the characteristics of the supercomputer used for the processes of lemmatization, labeling and entity identification as well as the library used to parallelize this procedure. The resources consumed in this process are also mentioned. Line 267. 

In Section 4, the authors did a deep analysis of the presence or absence of key terms. However, the reviewer feel not clear how the authors reached to the conclusion that “Irrespective of the interpretation, with the computational corpus ready, those specialists can focus on their analytic insights, instead of expending a very large amount of time and effort to the huge task of cross-reading all the laws.” Perhaps, more thorough discussion should be given. Also, the reviewer suggest adopt a second case to support the claim and show the function.

A. Section 4 was modified and an analysis with TF.IDF was added.

Reviewer 2 Report

Not every contribution to science needs to establish a sweeping new paradigm. Sometimes an article advances knowledge by straightforwardly applying known methods to a new problem. This article belongs to this category of "normal science." It deserves publication.

Converting a corpus of fundamental legal documents, especially in a civil law system such as Mexico's (where the text of a constitution and statutes based upon it are primary drivers of law), is an essential but often neglected task in natural language processing (NLP). This article elevates the usual methods for converting PDFs and parsing them into JSON format. It adds the critical step of part-of-speech tagging according to EAGLE standards, with very high accuracy rates.

I recommend publication after minor revisions. I have a few specific recommendations, all of which stem from this broad, overall suggestion: It is always difficult to strike the balance between explanatory clarity for newcomers and explanatory completeness for experts. At the same time, you do want to cover all ground without boring NLP veterans. There are just a few places where the authors can improve this admittedly difficult balance.

Section 4 is probably the one portion of the article that could stand considerable expansion and revision. For instance: The article observes that the standard deviation among the number of documents in which a semantic entity appears exceeds the mean. This is a regular occurrence in NLP. Indeed, it suggests that symmetrical, elliptical distributions are ill-suited for describing word and document frequency. There is an entire body of work, canonical within NLP, dating back to George Kinsley Zipf and known today as "Zipf's law." These are right-skewed distributions that make NLP more complex than many other mathematically similar tasks within data science.

Indeed, the fact that standard deviation exceeds the mean invites a fuller discussion of term frequency-inverse document frequency methods in NLP (TF-IDF). There is much more that contemporary NLP can do with a well processed corpus of legal documents. But TF-IDF is an essential first step, and the authors should guide their readers through the application of this basic NLP tool to their corpus.

Two other suggestions border on the trivial, but they are worth noting nonetheless. First, Figure 1 is tiny. It is worth expanding the text in each box and outlining the content in this less-than-helpful graphic so that readers who are not fully familiar with NLP and EAGLE can follow each of the steps.

Second, the abbreviation poder_judicial_de_la_federación should include the (Spanish) word judicial. Otherwise the abbreviation just means "federal power" in the abstract, and not the Federal Judiciary. ¿Tengo razón, no?

These suggestions are just that — suggestions. The article's worthiness does not hinge on the authors' agreement with one reviewer's suggestions. The editors should proceed confidently to an acceptance of this very well conceived and very competently executed work of normal science. 

Author Response

Section 4 is probably the one portion of the article that could stand considerable expansion and revision. For instance: The article observes that the standard deviation among the number of documents in which a semantic entity appears exceeds the mean. This is a regular occurrence in NLP. Indeed, it suggests that symmetrical, elliptical distributions are ill-suited for describing word and document frequency. There is an entire body of work, canonical within NLP, dating back to George Kinsley Zipf and known today as "Zipf's law." These are right-skewed distributions that make NLP more complex than many other mathematically similar tasks within data science. 

A: Section 4 was modified and an analysis with TF.IDF was added. 

Indeed, the fact that standard deviation exceeds the mean invites a fuller discussion of term frequency-inverse document frequency methods in NLP (TF-IDF). There is much more that contemporary NLP can do with a well processed corpus of legal documents. But TF-IDF is an essential first step, and the authors should guide their readers through the application of this basic NLP tool to their corpus. 

A: Section 4.3 was added 

Two other suggestions border on the trivial, but they are worth noting, nonetheless. First, Figure 1 is tiny. It is worth expanding the text in each box and outlining the content in this less-than-helpful graphic so that readers who are not fully familiar with NLP and EAGLE can follow each of the steps. 

A: The size of the figures and tables was modified for better viewing. Page 3,4, 7 and 8. 

Second, the abbreviation poder_judicial_de_la_federación should include the (Spanish) word judicial. Otherwise the abbreviation just means "federal power" in the abstract, and not the Federal Judiciary. ¿Tengo razón, no? 

A: The abbreviations mentioned above have been modified., Line 190 

Reviewer 3 Report

Contribution of the paper:

The authors have built a corpus of Mexican legal texts, taken form official sources in PDF form, and developed a database in JSON format, with a coarse structure for each document, including not only the text, but also term frequencies, lemmas, Part-of-speech (POS) tags and Named Entites (NEs).

Pros:

* The paper is readable and easy to understand.

* The English is good, though not perfect.

* The subject of this work matches the scope of this journal.

* I have not seen improper or unsupperted claims, or obvious methodological errors, though there is not a lot material in this paper to draw errors from (see below).

Cons:

* I fail to see the entity or relevance of this work. The authors extracted plain text from PDF files using available software, developed a coarse estructure for the legal documents, and used existing tools to perform POS tagging, lemmatization, and Named Entity Recognition (NER). The only relevant contributions would be the JSON structure developed by the authors and the python code used to clean the extracted text.

* I assume that the processing of the extracted text into the JSON structure developed by the authors was done automatically, but I fail to see where in the text of this paper is this process described or evaluated, so I could be mistaken.

* The authors perform automatic POS tagging, lemmatization, and NER, without manual correction. There is no evaluation on how well the tools chosen for these tasks perform, which could have been conducted by choosing a subset of documents or sentences and have them checked by human evaluators. I think that kind of testing would have been useful for researchers, since, if I am not mistaken, the Spanish models in both spaCy and FreeLing are trained on a corpus (AnCora) composed mainly of newspaper texts in current European Spanish. I would expect some degradation of their performances when applied on Mexican legal documents spanning several decades.

* The authors use spaCy for NER and FreeLing for POS tagging and lemmatization. This is strange, since spaCy has components for the latter two tasks in Spanish. Why was not spaCy used? Given that its POS tagging module annotates with the same coarse POS tags present in the Universal Dependencies treebanks, I think it could be useful to add its output to the more detailed EAGLES tags from FreeLing. In addition, dependency parsing could have been added, in the interest of further semantic analysis of the texts.

* The authors do not say how the plain text is processed and tokenized previously to NER and POS tagging. Given that, in Figure 1, NER is preformed before POS tagging, I assume that they are using spaCy for preprocessing the text as well as for NER. But I have not seen it clearly stated anywhere in the text.

* Which tool (spaCy of FreeLing) is used for lemmatization? Again, I have not seen it clearly stated in the text.

* The authors do not give information about the kinds of Named Entities that have been identified (Organizations, Persons, etc), or their percentages on the total number of Named Entities. That information could be useful for researchers.

* Regarding the size of the corpus, only document counts are given. I think word/sentence counts should be provided, along with statistics on average document length, number of titles/chapters/articles per document, etc.

* In general, the font size in the figures is a little too small. In addition, the grey tones used in the descriptions of the JSON structure is too light. That makes reading difficult. A more vivid color should be used.

* There are some errors, typos and not-so-good English in the text. Without being exhaustive:

- "Mexican Constitution" is repeatedly used, instead of "The Mexican constitution", which I think is the correct form.

- In section 2, paragraph 3: "two Natural Language Processing task were applied". It should be "tasks".

- The authors repeatedly use the acronym EAGLE for the standard used to develop the tagset used by FreeLing. The correct acronym is EAGLES (Expert Advisory Group on Language Engineering Standards).

- I would use spaCy, instead of Spacy, as the name of the Natural Language Processing API used for NER (and possibly for preprocessing and tokenization).

- The reference for FreeLing is missing, as well as the reference for the EAGLES tagset.

- In page 3, paragraph 1 it is stated that "Documents were collected from the official site [12] in PDF format. Whose official site?

- In subsection 2.2, paragraph 1: "Python algorithms were implemented to identifies and treat:...". It should be "identify". In addition, "scripts" or "programs" would probably have been better than "algorithms", and "deal with" better than "treat".

- In section 2.3, when the contents stored in the JSON structure are enumerated, "Grammar tagging" should be replaced by "Part-of-speech tagging", "POS tagging" or "Morphosyntactic tagging".

- In section 5, there is a typo in "nos specialized".

- The types of publications in references [1] and [11] are in Portugese. They should be translated to English.

Author Response

* I fail to see the entity or relevance of this work. The authors extracted plain text from PDF files using available software, developed a coarse estructure for the legal documents, and used existing tools to perform POS tagging, lemmatization, and Named Entity Recognition (NER). The only relevant contributions would be the JSON structure developed by the authors and the python code used to clean the extracted text. 

A: Yes, the JSON model of representation is one of the contributions of this work, the second contribution is properly the methodology implemented for the creation of the corpus and as final product the set of documents structured and ready for computational use show the relevance of this work. The article advances knowledge by directly applying known methods to a new problem. 

* I assume that the processing of the extracted text into the JSON structure developed by the authors was done automatically, but I fail to see where in the text of this paper is this process described or evaluated, so I could be mistaken. 

A: The proposed document storage structure was analyzed and validated by NLP and legislative analysis experts. The storage of the different types of documents in the same structure without loss of information supports the proposed mode. (Lines 138, 116, 201, 242) 

* The authors perform automatic POS tagging, lemmatization, and NER, without manual correction. There is no evaluation on how well the tools chosen for these tasks perform, which could have been conducted by choosing a subset of documents or sentences and have them checked by human evaluators. I think that kind of testing would have been useful for researchers, since, if I am not mistaken, the Spanish models in both spaCy and FreeLing are trained on a corpus (AnCora) composed mainly of newspaper texts in current European Spanish. I would expect some degradation of their performances when applied on Mexican legal documents spanning several decades. 

A: We used spaCy for NER and Freeling for lemmatization and PoS tagging. Characteristics about them were added in lines 98, 178, 222. 

* The authors use spaCy for NER and FreeLing for POS tagging and lemmatization. This is strange, since spaCy has components for the latter two tasks in Spanish. Why was not spaCy used? Given that its POS tagging module annotates with the same coarse POS tags present in the Universal Dependencies treebanks, I think it could be useful to add its output to the more detailed EAGLES tags from FreeLing. In addition, dependency parsing could have been added, in the interest of further semantic analysis of the texts. 

A: spaCy library and Freeling reach an accuracy of 97% according to references. We used Freeling because of the execution time. (paragraph in line 222) 

* The authors do not say how the plain text is processed and tokenized previously to NER and POS tagging. Given that, in Figure 1, NER is preformed before POS tagging, I assume that they are using spaCy for preprocessing the text as well as for NER. But I have not seen it clearly stated anywhere in the text. 

A: Two tools were used in the preprocessing of the document, stemming and tagging were done in Freeling. For the NER task, the spaCy library was implemented. PoS was performed after NER. And TF was performed before and after NER. We change some descriptions in Section 3 for a better description.  

* Which tool (spaCy of FreeLing) is used for lemmatization? Again, I have not seen it clearly stated in the text. 

A: Freeling was used for lemmatization. Line 210. 

* The authors do not give information about the kinds of Named Entities that have been identified (Organizations, Persons, etc), or their percentages on the total number of Named Entities. That information could be useful for researchers. 

A:The statistics regarding NER were made on a subset of documents specialized in environmental legislation. The numbers can be seen on line 201. 

* Regarding the size of the corpus, only document counts are given. I think word/sentence counts should be provided, along with statistics on average document length, number of titles/chapters/articles per document, etc. 

A: Only statistics related to articles per document were added. Line 308. 

* In general, the font size in the figures is a little too small. In addition, the grey tones used in the descriptions of the JSON structure is too light. That makes reading difficult. A more vivid color should be used. 

A: The images were modified. 

* There are some errors, typos and not-so-good English in the text. Without being exhaustive: 

- "Mexican Constitution" is repeatedly used, instead of "The Mexican constitution", which I think is the correct form. 

A: We change it to the correct form 

- In section 2, paragraph 3: "two Natural Language Processing task were applied". It should be "tasks". 

A: We change it to the correct form (line 78) 

- The authors repeatedly use the acronym EAGLE for the standard used to develop the tagset used by FreeLing. The correct acronym is EAGLES (Expert Advisory Group on Language Engineering Standards). 

A: We modified it. 

- I would use spaCy, instead of Spacy, as the name of the Natural Language Processing API used for NER (and possibly for preprocessing and tokenization). 

A: This recommendation was addressed throughout the document. 

- The reference for FreeLing is missing, as well as the reference for the EAGLES tagset. 

A: References were included. Lines 174, 230 

- In page 3, paragraph 1 it is stated that "Documents were collected from the official site [12] in PDF format. Whose official site? 

A: References were included. Line 95 

- In subsection 2.2, paragraph 1: "Python algorithms were implemented to identifies and treat:...". It should be "identify". In addition, "scripts" or "programs" would probably have been better than "algorithms", and "deal with" better than "treat". 

A: These recommendations were served on the line 108 

- In section 2.3, when the contents stored in the JSON structure are enumerated, "Grammar tagging" should be replaced by "Part-of-speech tagging", "POS tagging" or "Morphosyntactic tagging". 

A: These recommendations were served on Figures of Section 2, and on the line 262 

- In section 5, there is a typo in "nos specialized". 

A: These recommendations were served on the line 379. 

- The types of publications in references [1] and [11] are in Portugese. They should be translated to English. 

A: These recommendations were served on References section 

Round 2

Reviewer 1 Report

The authors have addressed my questions.

Author Response

Missing references were added. Some statistics and validations were incorporated in sections 2.4 and 2.5.

Reviewer 3 Report

I see that most of my concerns have bee addressed, but there are still some issues that should be corrected:

a) I still do not see word/token counts for the whole corpus.

b) In section 2.4, the authors' conducted a study of the types of NER detected in a subset of the corpus (actually as a response to one of my concerns) but I do not see the number of NEs detected. In addition while they report some of the errors made by the NER module, they do not report the accuracy (or error rate) observed by their manual validation.

c) As in b), the authors report the results of a manual validation of the POS Tagger used in section 2.5, but they do not report the accuray (or error rate) observed, or the size (in words/tokens) of the document used in this validation.

d) The English used is still far from perfect, and should be corrected by a someone fluent in that language, or, better, a native speaker.

This is not a correction. Just a suggestion for the future improvement of the corpus. Regarding NER, both FreeLing and spaCy only seem to check if a sequence of contiguous words is a NE, without tagging the type of NE found (places, organizations, people, etc). Since there has been plenty of work in including entity types in NE tags (usually following a IOB2 scheme) it would be interesting to find a NER tool trained in this kind of more fine-grained recognition for Spanish, and add the entity tipe for each NE identified.

Author Response

a) I still don't see word/token counts for the entire corpus.
A. It was incorporated in Section 4.1

b) In section 2.4, the authors have conducted a survey of the types of NERs detected in a subset of the corpus (actually, in response to one of my concerns), but I do not see the number of NEs detected. Also, although they report some of the errors made by the NER module, they do not report the accuracy (or error rate) observed by their manual validation. 
A. Incorporated in the sixth paragraph of Section 2.4.

c) As in b), the authors report the results of a manual validation of the POS tagger used in section 2.5, but do not report the accuracy (or error rate) observed, nor the size (in words/tokens) of the document used in this validation.
A. Incorporated in Section 2.5, sixth paragraph

d) The English used is far from perfect, and should be corrected by someone fluent in that language, or, better, by a native speaker.
A. It was reviewed in general and some changes were made.